# Clinical and Morphological Features of ER-Positive HER2-Negative Breast Tumors with PIK3CA Mutations in Russian Patients

**DOI:** 10.3390/cancers17111833

**Published:** 2025-05-30

**Authors:** Tatyana N. Sokolova, Grigory A. Yanus, Svetlana N. Aleksakhina, Yana V. Belysheva, Aleksandra P. Chernyakova, Yulia S. Zharnakova, Alisa S. Nikitina, Tatyana M. Stebneva, Aleksandr S. Martianov, Alla Yu. Goryainova, Mark I. Gluzman, Rashida V. Orlova, Anastasiya I. Stukan’, Alena V. Zyuzyukina, Ruslan A. Zukov, Polina R. Korzun, Jeyla O. Binnatova, Anastasia S. Abuzova, Yulia N. Murunova, Aleksandr V. Sultanbaev, Elena N. Vorobeva, Leonid M. Mikhaevich, Victoria N. Pyliv, Anna N. Lysenko, Zarema K. Khachmamuk, Andrey E. Kozlov, Sergey Yu. Bakharev, Shagen G. Parsyan, Elena I. Rossokha, Leri D. Osidze, Irina S. Shumskaya, Anna V. Agaeva, Tatyana A. Kasmynina, Veronika V. Klimenko, Kamila T. Akhmetgareeva, Almira A. Vakhitova, Madina D. Chakhkieva, Vadim N. Dmitriev, Yana I. Bakshun, Alexey E. Vasiliev, Dunya D. Gasimly, Nadezhda A. Kravchenko, Dmitriy A. Maksimov, Alfia I. Nesterova, Ineza O. Sharvashidze, Christina Kh. Gadzaova, Galina G. Rakhmankulova, Zaur M. Khamgokov, Irina K. Amirkhanova, Ludmila V. Bembeeva, Vladimir I. Vladimirov, Oleg L. Petrenko, Natalia G. Ruskova, Ekaterina L. Serikova, Ksenia S. Subbotina, Svetlana A. Tkachenko, Victor L. Chang, Sanal P. Erdniev, Victoria S. Barbara, Anna V. Vasilevskaya, Yulia V. Mikheeva, Natalia O. Popova, Anastasia V. Fateeva, Denis Yu. Yukalchuk, Anna A. Grechkina, Khedi S. Musayeva, Svetlana V. Odintsova, Petimat I. Khabibulaeva, Alina G. Khlobystina, Kseniya A. Shvaiko, Elena A. Basova, Irina A. Bogomolova, Marina B. Bolieva, Viktor E. Goldberg, Marianna V. Kibisheva, Konstantin V. Menshikov, Dmitriy V. Ryazanov, Yana A. Udalova, Aleksandr V. Shkradyuk, Idris M. Khabriev, Dmitriy V. Kirtbaya, Alexey M. Degtyarev, Aleksandr A. Epkhiev, Yana A. Tyugina, Mirza A. Murachuev, Alena S. Stelmakh, Aglaya G. Iyevleva, Evgeny N. Imyanitov

**Affiliations:** 1National Medical Research Center of Oncology named after N.N. Petrov of MOH of Russia, Saint Petersburg 197758, Russia; stretanya@yandex.ru (T.N.S.); octavedoctor@yandex.ru (G.A.Y.); abyshevasv@gmail.com (S.N.A.); belysheva.yana24@gmail.com (Y.V.B.); aleksandr.s.martianov@gmail.com (A.S.M.); evgeny@imyanitov.spb.ru (E.N.I.); 2Saint-Petersburg State Pediatric Medical University, Saint Petersburg 194100, Russia; 3State Budgetary Healthcare Organization “Clinical Oncology Dispensary No.1” of the Ministry of Healthcare of Krasnodar Region, Krasnodar 350040, Russia; 4Saint Petersburg State University, Saint Petersburg 199034, Russia; 5State Budgetary Healthcare Institution “City Clinical Oncological Dispensary”, Saint Petersburg 197022, Russia; 6Federal State Budgetary Educational Institution of Higher Education “Prof. V.F. Voino-Yasenetsky Krasnoyarsk State Medical University” of MOH of Russia, Krasnoyarsk 660022, Russia; 7State Budgetary Healthcare Institution “Krasnoyarsk Regional Clinical Oncological Dispensary named after A.I. Kryzhanovsky”, Krasnoyarsk 660113, Russia; 8Budgetary Healthcare Institution of the Khanty-Mansiysk Autonomous District “Surgut Regional Clinical Hospital”, Surgut 628408, Russia; 9State Autonomous Healthcare Institution “Republican Clinical Oncological Dispensary” of MOH of the Republic of Bashkortostan, Ufa 450054, Russia; 10State Autonomous Healthcare Institution “Chelyabinsk Regional Clinical Center of Oncology and Nuclear Medicine”, Chelyabinsk 454087, Russia; 11Budgetary Institution of the Khanty-Mansiysk Autonomous District-Yugra “District Clinical Hospital”, Khanty-Mansiysk 628012, Russia; 12State Budgetary Healthcare Institution of the Stavropol Region “Stavropol Regional Oncological Center”, Stavropol 355047, Russia; 13State Healthcare Institution “Tula Regional Oncological Dispensary”, Tula 300039, Russia; 14Regional State Budgetary Healthcare Institution “Altai Regional Oncological Dispensary”, Barnaul 656045, Russia; 15 State Autonomous Healthcare Institution of Nizhny Novgorod Region “Nizhny Novgorod Regional Clinical Oncological Dispensary”, Nizhny Novgorod 603126, Russia; 16 State Budgetary Healthcare Institution of the Arkhangelsk Region “Arkhangelsk Clinical Oncological Dispensary”, Arkhangelsk 163045, Russia; 17 Regional State Budgetary Healthcare Institution “Oncological Dispensary”, Birobidzhan 679000, Russia; 18Bashkir State Medical University, Ufa 45008, Russia; 19State Budgetary Institution “Republican Oncological Dispensary named after G.M. Vedzizhev”, Plievo, Ingushetia 386124, Russia; 20Regional State Budgetary Healthcare Institution “Belgorod Regional Oncological Dispensary”, Belgorod 308010, Russia; 21Regional State Budgetary Healthcare Institution “Kostroma Clinical Oncological Dispensary”, Kostroma 156005, Russia; 22State Autonomous Healthcare Institution “Orenburg Regional Clinical Oncological Dispensary”, Orenburg 460021, Russia; 23State Budgetary Healthcare Institution “Regional Oncological Dispensary”, Irkutsk 664044, Russia; 24State Budgetary Healthcare Institution “Tver Regional Clinical Oncological Dispensary”, Tver 170008, Russia; 25Federal State Autonomous Educational Institution of Higher Education “Kazan (Volga Region) Federal University”, Kazan 420008, Russia; 26State Budgetary Healthcare Institution “Republican Oncological Dispensary”, Vladikavkaz 362002, Russia; 27State Budgetary Healthcare Institution of MOH of Kabardino-Balkarian Republic “Republican Oncological Dispensary”, Nalchik 360051, Russia; 28Budgetary Healthcare Institution of the Vologda Region “Vologda Regional Oncological Dispensary”, Vologda 160012, Russia; 29State Budgetary Healthcare Institution of the Stavropol Region “Pyatigorsk Interdistrict Oncological Dispensary”, Pyatigorsk 357502, Russia; 30State Budgetary Healthcare Institution “Regional Clinical Hospital of the Kaliningrad Region”, Kaliningrad 236016, Russia; 31State Budgetary Healthcare Institution of the Republic of Karelia “Republican Oncological Dispensary”, Petrozavodsk 185002, Russia; 32State Budgetary Healthcare Institution of the Kaluga Region “Kaluga Regional Clinical Oncological Dispensary”, Kaluga 248007, Russia; 33State Budgetary Healthcare Institution “Tambov Regional Oncological Clinical Dispensary”, Tambov 392000, Russia; 34State Budgetary Healthcare Institution of the Moscow Region “Moscow Regional Oncological Dispensary”, Balashikha 143900, Russia; 35Saint Petersburg State Budgetary Healthcare Institution “St. Luke Clinical Hospital”, Saint Petersburg 194044, Russia; 36Tomsk National Research Medical Center of Russian Academy of Sciences, Tomsk 634009, Russia; 37State Budgetary Healthcare Institution “Primorsky Regional Oncological Dispensary”, Vladivostok 690105, Russia; 38State Budgetary Institution “Republican Oncological Dispensary”, Grozny 364029, Russia; 39EuroCityClinic LLC, Saint Petersburg 197022, Russia; 40State Budgetary Healthcare Institution “Volgograd Regional Clinical Oncological Dispensary”, Volgograd 400117, Russia; 41 Federal State Budgetary Institution “Federal Scientific Clinical Center for Medical Radiology and Oncology” of FMBA of Russia, Dimitrovgrad 433507, Russia; 42State Budgetary Institution of the Ryazan Region “Regional Clinical Oncological Dispensary”, Ryazan 390011, Russia; 43 Taganrog Branch of the State Budgetary Institution of the Rostov Region “Oncological Dispensary”, Taganrog 347910, Russia; 44State Budgetary Healthcare Institution of the Republic of Crimea “Crimean Republican Oncological Clinical Dispensary named after V.M. Efetov”, Simferopol 295007, Russia; 45State Budgetary Institution of the Krasnodar Region “Oncological Dispensary No.2” of MOH of Krasnodar Region, Sochi 354057, Russia; 46Regional Budgetary Healthcare Institution “Ivanovo Regional Oncological Dispensary”, Ivanovo 153040, Russia; 47State Budgetary Institution of the Republic of Dagestan “Republican Oncology Center”, Makhachkala 367000, Russia; 48State Budgetary Healthcare Institution “Murmansk Regional Oncology Center”, Murmansk 183038, Russia

**Keywords:** hormone receptor-positive (HR+)/HER2-negative (HER2−) breast cancer, PIK3CA, mutation

## Abstract

Mutations in *PIK3CA* occur in approximately one-third of hormone receptor-positive (HR+)/HER2-negative (HER2−) breast cancer (BC) and represent a target for specific inhibitors. The current study established the frequency and spectrum of *PIK3CA* alterations in HR+/HER2− BC from Russian patients. We conclude that the overall prevalence and characteristics of *PIK3CA*-positive cases are consistent with previously published data and do not depend on the patient’s ethnic background. We also show that *PIK3CA* testing should not be limited to hot-spot variants, as commercially available kits may miss up to 12% of *PIK3CA*-mutated cases.

## 1. Introduction

Mutations affecting *PIK3CA*, the gene encoding the catalytic subunit of p110-alpha kinase PI3K, represent one of the most common genetic driver events in human tumors [1]. The attempts to develop inhibitors of the PI3K/AKT/mTOR signaling cascade emerged some time ago, but it was not until recently that the balance between clinical efficacy and drug tolerability could be achieved. Monotherapy of solid neoplasms with PI3K inhibitors proved ineffective, however, several compounds have already been approved for the treatment of *PIK3CA*-mutated hormone-receptor (HR)-positive/HER2-negative breast cancer (BC) in combination with other agents. Combined use of a selective inhibitor of the PI3K-alpha alpelisib and estrogen degrader fulvestrant received approval for patients progressing on or after prior endocrine-based therapy [2,3]. Similar approval was granted to AKT inhibitor capivasertib; however, in addition to *PIK3CA* mutations, the indication of this drug includes nucleotide sequence alterations in *AKT1* and *PTEN* genes [4]. Another AKT inhibitor, ipatasertib, is currently under investigation in breast cancer [5]. Inavolisib has the ability to drive the degradation of mutated p110-alpha while sparing the wild-type isoform of this protein; it has been approved for clinical use in combination with fulvestrant and CDK4/6 inhibitor palbociclib [6]. Several other PI3K inhibitors have failed to achieve a significant improvement in the progression-free survival or objective response rate, and their use has been associated with considerable toxicity (e.g., buparlisib, pictilisib, and taselisib). A number of novel compounds, including inhibitors that selectively target mutant PIK3CA isoforms, have shown promising results in preclinical and early-phase clinical studies [5]. There are also ongoing trials utilizing combinations of PI3K inhibitors with other drugs and involving cancer types other than BC [7,8,9,10].

*PIK3CA* mutations affect a variety of biological properties of tumor cells, including some essential metabolic features, profiles of secreted immunocytokines, regulation of the cell cycle, formation of tolerance to polyploidy and chromosomal instability, etc. [11,12,13]. Activating *PIK3CA* lesions were shown to have a prognostic value in breast cancer, however, their role may differ in early- and late-stage tumors. In particular, *PIK3CA* mutations appear to be associated with a favorable prognosis in early BC, while in metastatic HR+ and/or HER2+ carcinomas, they may be enriched in cases with unfavorable disease course and resistance to standard therapy schemes [14,15,16]. The presence of *PIK3CA* alterations negatively affects the results of targeted therapy in HER2-positive BC [17,18,19,20].

There are several hotspot *PIK3CA* regions most frequently affected by somatic alterations. About 70% of identified mutations are located in codons 542, 545 (exon 9, helical domain) and 1047 (exon 20, kinase domain). Substitutions in the helical domain initiate transmission of a signal through the RAS-MAPK molecular cascade independently of the regulatory PI3K subunit, p85. In contrast, kinase domain mutations are associated with the requirement for interaction with p85, but the activation of signaling does not depend on RAS-GTP binding [21]. Recently, a phenomenon of multiple (usually two) co-occurring *PIK3CA* mutations was described in breast cancer and other tumor types: these mutations occur mainly in cis, are mutually exclusive with other driver genetic alterations, and confer increased sensitivity to PI3K inhibition [22,23]. *PIK3CA* H1047R substitution was shown to be associated with a low proportion of complete responses to anthracycline and taxane therapy in triple-negative breast cancer, and with resistance to anti-HER2 therapy in HER2-positive tumors [24,25]. Mutations in the helical domain occur more frequently in well-differentiated neoplasms of the luminal A expression subtype, as well as in lobular BC [26,27]. Despite the functional differences, the predictive role of exon 9 and exon 20 mutations in relation to alpelisib seems to be similar [2]. Less frequently, somatic mutations are observed in the C2 domain of PI3K (exon 7) and in the other gene parts. The clinical and biological significance of atypical mutations in the helical and kinase domains, as well as of rare alterations in *PIK3CA*, have not yet been fully characterized [28], although the evidence for correlation of several types of rare variants with clinical benefit from PI3K inhibition continues to emerge [29,30].

There is some evidence that the occurrence of *PIK3CA* mutations in breast cancer may have interethnic differences [31,32,33]. The aim of the current study was to analyze the frequency, spectrum, and clinical associations of *PIK3CA* lesions in Russian patients with HR+/HER2− breast cancer.

## 2. Materials and Methods

The study included 1872 patients with ER+/HER2− breast cancer diagnosed between 1980 and 2022 and treated in more than 50 cities of Russia. All patients were referred to *PIK3CA* mutation analysis in 2020–2023, with about 85% of them suffering from the recurrent or metastatic disease at the time of genetic testing. Approximately half of patients underwent *PIK3CA* testing in the first 4 years after the disease manifestation. The study cohort consisted of 1861 women and 11 men with breast tumors. The median age of the patients was 51 years (range: 20–88 years). DNA for genetic analysis was extracted from formalin-fixed paraffin-embedded (FFPE) primary tumor tissues (≈70%) or FFPE material from tumors obtained from tumor recurrence/metastasis (≈30%). Testing for *PIK3CA* mutations was performed by a combination of high-resolution melting (HRM) analysis, allelic discrimination TaqMan PCR (AD-PCR), digital droplet PCR, and pyrosequencing. The presence of alterations in exons 2 (codons 64–117), 7 (codons 418–455), 10 (codons 526–552), 20 (codons 1027–1057) was determined by HRM of PCR products; mutations G118D, N345K, E726K were investigated by AD-PCR. Cases showing abnormal melting patterns were tested for hot-spot variants by the corresponding AD-PCR assays (exon 2: codons 81, 88, 108, 110, 111; exon 7: codons 420, 453; exon 10: codons 542, 545, 546; exon 20: codons 1043, 1044, 1047, 1049). Tumor samples with equivocal results were further analyzed by digital droplet PCR assays for the mutations E542K, E545K, H1047R, and H1047L. Cases with abnormal melting curves, which were negative for the hot-spot variants, were subsequently subjected to pyrosequencing. The list of primers and TaqMan probes, assay conditions, and utilized equipment is given in Appendix A.

Associations between PIK3CA alterations and clinicopathological parameters were estimated using the chi-squared test. The Mann–Whitney U test was used to compare age in patients with different PIK3CA status. Event-free survival (EFS) was determined as the interval between diagnosis and detection of disease recurrence or distant metastasis in stage I-III cases. Data on the time of disease progression were extracted from available medical records. Kaplan–Meier method and log-rank test were used to construct and compare survival curves to determine the parameters affecting EFS. The Cox proportional hazards regression model was used to assess the prognostic significance of multiple parameters simultaneously. Variables were entered into the Cox regression model using a stepwise method. *p* < 0.05 was considered statistically significant. Statistical analyses were performed using MedCalc software (Version 19).

## 3. Results

*PIK3CA* mutations were identified in 693/1872 (37.0%) patients (687/1861 (36.9%) female and 6/11 (54.5%) male BC cases). Forty-six out of 693 tumors (6.6%) harbored two *PIK3CA* alterations simultaneously (Appendix A). At least one mutation in exon 10 or 20 was identified in the majority of *PIK3CA*-positive BC (624/693, 90%). The most frequent substitution types were known major hotspots: p.H1047R in exon 20 (310/739, 42.0%), p.E545K (153/739, 20.7%), and p.E542K (84/739, 11.4%) in exon 10. These three alterations were identified in 542/693 (78.2%) *PIK3CA*-mutated cases. Several other missense variants occurred more than 10 times: p.H1047L (n = 30), p.N345K (n = 28), p.E726K (n = 23), p.C420R (n = 17), p.Q546K (n = 15), p.E453K (n = 12) (Figure 1). P.E726K substitution (exon 14) was predominantly observed in double-mutated cases (18/23, 78.3%). There was a trend towards a higher mutation detection rate in primary tumor samples than in metastatic material (39.0% vs. 34.3%, *p* = 0.107).

The median age at BC diagnosis was higher in *PIK3CA*-mutated cases (53.0 vs. 50.0 years, *p* = 0.0002); *PIK3CA* variants were detected in 43.9% of patients older than 60 years compared to 33.9% in younger individuals (*p* = 0.0002). *PIK3CA* alterations were associated with a number of favorable prognostic features: smaller tumor size, lower grade, Ki67 <20%, and positive PR status (*p* ≤ 0.005 for all comparisons) (Figure 2, Table 1). In our case series, about 85% of patients had metastatic disease at the time of *PIK3CA* testing. *PIK3CA* mutations were found less frequently in cases with multiple metastases than in patients with only one metastasis-affected zone (32.8% vs. 43.1%, *p* = 0.001). Mutations also tended to be underrepresented in patients with distant metastases to soft tissues, skin, and lymph nodes (Table 1).

Our study sample included BC patients residing in different parts of Russia. Women living in the North Caucasus belong mostly to non-Slavic ethnic groups (Kabardins, Ossetins, Chechens, Karachays, Avars, Ingush) and are generally characterized by earlier age at first delivery and a higher number of childbirths when compared to Western-style communities. For example, the total fertility rate in 2023 was 1.24 in North Western Russia and 1.72 in the North Caucasus [34]. The above factors significantly affect BC epidemiology: BC incidence per 100,000 women in the North Caucasus is among the lowest in Russia [35]. However, patients from this region had no specific features with regard to the spectrum and frequency of *PIK3CA* mutations. Other regions with predominantly non-Slavic populations, e.g., the Republics of Tatarstan and Bashkortostan, also had distribution of *PIK3CA* alterations similar to the general sample. Women residing in an extremely cold climate (e.g., Khanti-Mansi region) had essentially the *PIK3CA* mutation rate as patients living in Southern Russia (Table 2, Figure 2).

Median event-free survival (EFS) in stage I–III tumors was 3 years. In univariable analysis, longer EFS was associated with patients’ age less than 60 years, lower tumor stage, absence of nodal involvement, and smaller tumor size. *PIK3CA* mutations had no effect on EFS (Table 3, Figure 3). In Cox regression analysis, including age, T, and N categories, these variables retained statistical significance in relation to EFS.

We were able to obtain the data on sensitivity to adjuvant hormonal therapy in 413 BC cases: 331 patients received tamoxifen and 82 were treated by aromatase inhibitors (anastrozole or letrozole). Patients were categorized as having primary endocrine resistance (a relapse within the first two years of adjuvant therapy), secondary resistance (defined as disease progression detected after the first two years of adjuvant therapy or within a year after treatment completion), or sensitivity (disease relapse diagnosed in more than a year after completion of adjuvant treatment or later) [36]. *PIK3CA* mutation prevalence did not significantly differ between cases with sensitivity or resistance to aromatase inhibitors or tamoxifen (Table 4).

## 4. Discussion

This study analyzed a large sample of Russian patients with HR+/HER2− BC who were referred for *PIK3CA* genetic testing between 2020 and 2023. The spectrum of the mutations identified was generally consistent with the published data [37,38], although the incidence of *PIK3CA* alterations was lower (37%) compared to that in European patients with ER+/HER2− BC (7706/17,687, 44%) [33]. The most plausible explanation for these differences is the distinct age distribution in the patient series. Indeed, our BC group had a relatively low median age at diagnosis (51 years) compared to the other large BC cohorts (58–61 years) [33,39]. Importantly, in our study, the frequency of *PIK3CA* mutations in subjects older than 60 years approached 44%. Similar to previous observations, the three main hotspot variants (p.H1047R, p.E545K, p.E542K) constituted 78% of all *PIK3CA* mutations [40].

Double mutations account for 12–15% of all BC cases with *PIK3CA* alterations and confer increased benefit from alpelisib therapy [22,23]. Our cohort contained 6,6% of co-occurring *PIK3CA* alterations. Most of the double mutations (40/46, 87%) included variants in major hotspot codons E542, E545, and H1047. Substitution E726K was enriched in cases with multiple mutations: it was observed as a single variant in only 5/23 BC (21.7%) and was one of the co-occurring variants in 18/46 (39%) double-mutated cases. The second common minor mutation in double-mutated BC was E453K/Q, which is consistent with previous findings [22].

Currently, there are a number of commercially available PCR-based *PIK3CA* mutation kits aimed at detecting the most common missense variants (e.g., therascreen PIK3CA RGQ PCR Kit (QIAGEN Manchester, Manchester, UK; cobas PIK3CA Mutation Test (Roche Diagnostics, Mannheim, Germany); AmoyDx PIK3CA Mutation Detection Kit (Amoy Diagnostics, Xiamen, China); Idylla PIK3CA-AKT1 Mutation Assay (BioCartis Idylla, Mechelen, Belgium, etc.)). These kits detect 11–17 *PIK3CA* variants. Using our stepwise PCR-based approach, we identified 43 unique mutation types, including 39 missense variants and 4 indels. Twenty-five of these variants were identified more than once, 10 of which could not be targeted by any of the commercial assays mentioned. Notably, neither of the above commercial assays included testing for the E453K and E726K variants, which were detected 12 and 23 times, respectively, in our dataset. Compared to our approach, Cobas, Idylla, Therascreen, and AmoyDx PIK3CA assays would miss 38/693 (5.5%), 42/693 (6.1%), 83/693 (12%), and 83/693 (12%) *PIK3CA*-mutated cases, respectively.

Most of the identified PIK3CA alterations (37 out of 43 mutation types and 731 out of a total of 739 mutations) have been reported previously. However, we could not find any mention of several of the identified variants (p.P124S, p.H419_C420delinsR, p.H419_P421>QT, p.C420_P421del, p.N1044H, and p.H1048A) in the literature. There is experimental evidence of the activating role of the majority of the found alterations (30/43 mutation types; 723/739 individual mutations) [28,41,42,43,44,45,46]. Nevertheless, the functional significance of the newly identified mutation types remains to be evaluated.

*PIK3CA* mutation status may be discordant between primary and metastatic tumor lesions in up to 10% of BC, with most of the discrepancy attributed to the loss of *PIK3CA* in metastatic tissues [47]. In accordance with this, we observed a trend towards lower *PIK3CA* occurrence in metastatic material.

Accumulating evidence suggests that the impact of *PIK3CA* mutations on tumor behavior and prognosis depends on the BC subtype. In HR+/HER2− BC, *PIK3CA* has been related to improved clinical outcome in operable primary cases, and to poor prognosis in metastatic disease [14,15,48,49,50]. We could not demonstrate an association between *PIK3CA* alterations and event-free survival in stage I-III BC, although their presence was strongly associated with smaller tumor size, lower grade, Ki67<20%, and PR protein expression. We also found that *PIK3CA* variants were more prevalent in patients with lower metastatic burden (43% vs. 33%, *p* = 0.0001). Our findings on the clinical associations of *PIK3CA* alterations generally replicate the data from several previous studies involving at least 400–500 BC cases [15,42,51,52,53,54,55]. At the same time, in most of these works, authors did not distinguish between individual BC subtypes, and the associations were described for the entire study sample. Our study included a large number of cases and focused on a relatively homogenous category of BC, i.e., patients with ER+/HER2− tumors.

ER and PI3K/AKT/mTOR signaling pathways are characterized by a complex interplay in breast tumors, and their coordinated activation supports the survival of ER-positive BC cells [56]. It is generally accepted that the activated PI3K/AKT/mTOR molecular cascade is implicated in endocrine resistance [57,58]; however, *PIK3CA* mutations were not associated with reduced benefit from adjuvant endocrine therapy in most of the previous studies. In fact, they were not significantly associated with outcomes of adjuvant tamoxifen therapy [55,59,60,61] and showed conflicting associations in relation to aromatase inhibitors. Ramirez-Ardila et al. [60] demonstrated an increased benefit of aromatase inhibitors in *PIK3CA*-mutated cases, while a recent study reported an opposite effect [55]. However, the authors of the latter work failed to validate their findings in an independent cohort [62]. We did not find an enrichment of *PIK3CA* mutations in patients with primary or secondary resistance to adjuvant tamoxifen or aromatase inhibitors. It was suggested that *PIK3CA* alterations do not confer pronounced resistance to endocrine therapy because they render only weak activation of the canonical PI3K pathway and probably exert their oncogenic potential through alternative molecular mechanisms [59,63].

## 5. Conclusions

In conclusion, our results highlight the importance of expanded testing of the *PIK3CA* gene, not limited to hotspot mutations. Although *PIK3CA* alterations contribute to the pathogenesis of HR+/HER2− BC and represent a target for several novel drugs, they are not intrinsically associated with unfavorable features of this subtype of cancer disease.

## Figures and Tables

**Figure 1 cancers-17-01833-f001:**
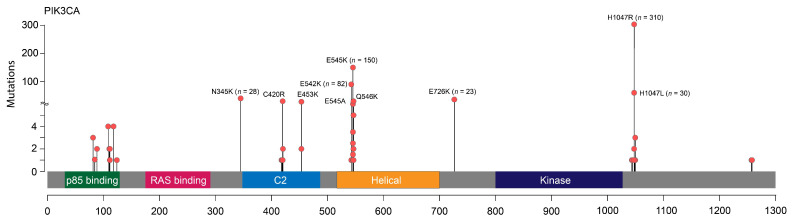
Schematic representation of the spectrum of *PIK3CA* mutations.

**Figure 2 cancers-17-01833-f002:**
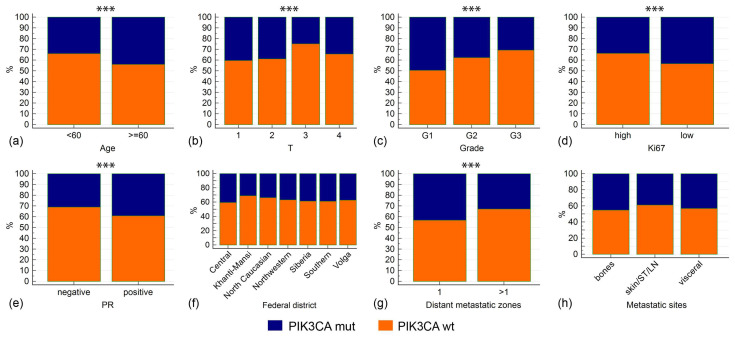
Associations between *PIK3CA* mutations and clinicopathological characteristics. Asterisks note the comparisons with *p* < 0.01. Clinicopathological characteristics include age at diagnosis (**a**), tumor size (**b**), tumor grade (**c**), Ki67 proliferation index (**d**), PR status (**e**), Federal district (**f**), number of metastatic zones (**g**), types of metastatic sites (**h**). In (**f**), only Federal districts with >100 analyzed cases are shown. In (**h**): ST—soft tissues, LN—lymph nodes.

**Figure 3 cancers-17-01833-f003:**
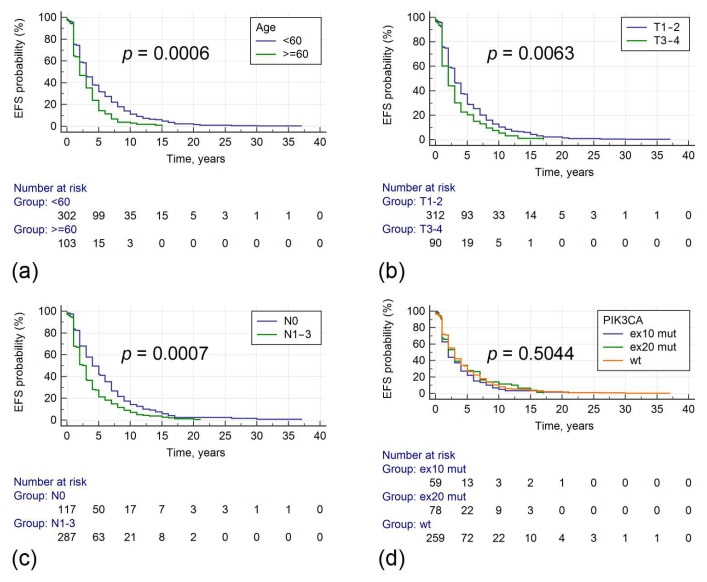
Associations between clinical parameters, *PIK3CA* mutations, and event-free survival. The evaluated parameters include age at diagnosis (**a**), tumor size (**b**), regional lymph node status (**c**), PIK3CA mutations (**d**).

**Table 1 cancers-17-01833-t001:** Clinical and pathological characteristics of the studied BC samples in relation to *PIK3CA* mutations.

Characteristics	PIK3CA WT (n = 1179)	PIK3CA MUT (n = 693)	*p*-Value (PIK3CA WT vs. MUT)	PIK3CA MUT (Exon 10, n = 257)	PIK3CA MUT (Exon 20, n = 323)	*p*-Value (Exon 10 vs. Exon 20 Mutations)
Age at diagnosis (mean ± SD), range	50.67 (±11.13); range: 20–83	52.77 (±11.55); range: 24–88)	0.0002	53.57 (±11.33); (range: 29–88)	52.07 (±11.22); (range: 24–85)	0.110
Histologic type						
Invasive cancer of no special type (NST) (n = 1448)	903 (86.2%)	545 (86.1%)	0.909	191 (83.0%)	267 (89.0%)	0.423
Invasive lobular cancer (ILC) (n = 156)	96 (9.2%)	60 (9.5%)	(NST vs. ILC)	23 (10.0%)	24 (8.0%)	(NST vs. ILC)
Other (n = 77)	49 (4.7%)	28 (4.4%)		16 (7.0%)	9 (3.0%)	
ND (n = 191)						
Primary tumor size (T)						
T1 (n = 386)	230 (21.1%)	156 (24.3%)	0.005	60 (25.0%)	77 (26.1%)	0.114
T2 (n = 809)	495 (45.4%)	314 (48.8%)	(T1–T2 vs. T3–T4)	109 (45.4%)	149 (50.5%)	(T1–T2 vs. T3–T4)
T3 (n = 125)	94 (8.6%)	31 (4.8%)		13 (5.4%)	11 (3.7%)	
T4 (n = 414)	272 (24.9%)	142 (22.1%)		58 (24.2%)	58 (19.7%)	
Other/ND (n = 138)						
Lymph node involvement (N)						
N0 (n = 485)	302 (27.9%)	183 (29.1%)	0.780	63 (27.0%)	91 (31.4%)	0.713
N1 (n = 588)	371 (34.3%)	217 (34.5%)		86 (36.9%)	97 (33.4%)	
N2 (n = 323)	212 (19.6%)	111 (17.6%)		42 (18.0%)	49 (16.9%)	
N3 (n = 314)	196 (18.1%)	118 (18.8%)		42 (18.0%)	53 (18.3%)	
ND (n = 162)						
Distant metastasis (M)						
M0 (n = 1240)	769 (79.5%)	471 (81.9%)	0.253	174 (81.7%)	215 (82.4%)	0.847
M1 (n = 302)	198 (20.5%)	104 (18.1%)		39 (18.3%)	46 (17.6%)	
ND (n = 330)						
Bilateral/unilateral disease						
Unilateral BC (n = 1785)	1118 (94.8%)	667 (96.2%)	0.194	247 (96.1%)	307 (95.3%)	0.606
Bilateral BC (n = 87)	61 (5.2%)	26 (3.8%)		10 (3.9%)	15 (4.7%)	
Stage						
I (n = 184)	115 (10.6%)	69 (11.0%)	0.536	22 (9.6%)	38 (13.1%)	0.557
II (n = 576)	353 (32.7%)	223 (35.6%)		80 (34.8%)	104 (36.0%)	
III (n = 619)	396 (36.6%)	223 (35.6%)		86 (37.4%)	97 (33.6%)	
IV (n = 329)	217 (20.1%)	112 (17.9%)		42 (18.3%)	50 (17.3%)	
ND (n = 164)						
Grade						
G1 (n = 99)	50 (7.6%)	49 (12.4%)	0.005	14 (9.7%)	26 (14.0%)	0.673
G2 (n = 726)	452 (68.4%)	274 (69.5%)		103 (71.5%)	124 (66.7%)	
G3 (n = 230)	159 (24.1%)	71 (18.0%)		27 (18.8%)	36 (19.4%)	
ND (n = 817)						
PR status						
Negative (n = 485)	335 (28.9%)	150 (22.2%)	0.002	59 (23.5%)	69 (21.7%)	0.608
Positive (n = 1350)	823 (71.1%)	527 (77.8%)		192 (76.5%)	249 (78.3%)	
ND (n = 37)						
Ki67						
Low (<20%) (n = 563)	319 (32.8%)	244 (42.4%)	0.0001	79 (36.7%)	126 (47.2%)	0.021
High (≥20%) (n = 984)	653 (67.2%)	331 (57.6%)		136 (63.3%)	141 (52.8%)	
ND (n = 325)						
Number of metastatic sites						
N = 1 (n = 531)	302 (32.9%)	229 (43.2%)	0.0001	91 (44.8%)	100 (42.6%)	0.633
N > 1 (n = 917)	616 (67.1%)	301 (56.8%)		112 (55.2%)	135 (57.4%)	
ND (n = 424)						
Distant metastases						
Bones						
Yes (n = 874)	548 (59.5%)	326 (61.2%)	0.389	115 (56.7%)	152 (64.1%)	0.110
No (n = 580)	373 (40.5%)	207 (38.8%)		88 (43.3%)	85 (35.9%)	
Brain						
Yes (n = 46)	34 (3.7%)	12 (2.3%)	0.133	5 (2.5%)	5 (2.1%)	0.810
No (n = 1405)	886 (96.3%)	519 (97.7%)		198 (97.5%)	231 (97.9%)	
Liver						
Yes (n = 453)	291(31.6%)	162 (30.4%)	0.240	67 (33.0%)	71 (30.0%)	0.493
No (n = 1000)	629 (68.4%)	371 (69.6%)		136 (67.0%)	166 (70.0%)	
Lung						
Yes (n = 457)	295 (31.0%)	162 (29.3%)	0.474	55 (26.7%)	81 (32.3%)	0.218
No (n = 1046)	656 (69.0%)	390 (70.7%)		151 (73.3%)	170 (67.7%)	
Lymph nodes						
Yes (n = 579)	383 (41.5%)	196 (36.8%)	0.084	83 (40.7%)	80 (33.9%)	0.142
No (n = 877)	541 (58.5%)	336 (63.2%)		121 (59.3%)	156 (66.1%)	
Peritoneum						
Yes (n = 38)	28 (3.0%)	10 (1.9%)	0.181	4 (2.0%)	5 (2.1%)	0.918
No (n = 1414)	892 (97.0%)	522 (98.1%)		199 (98.0%)	232 (97.9%)	
Pleura						
Yes (n = 160)	100 (10.9%)	60 (11.3%)	0.801	18 (8.9%)	30 (12.7%)	0.199
No (n = 1291)	820 (89.1%)	471 (88.7%)		185 (91.1%)	206 (87.3%)	
Skin						
Yes (n = 118)	84 (9.1%)	34 (6.4%)	0.069	10 (4.9%)	18 (7.6%)	0.249
No (n = 1335)	838 (90.9%)	497 (93.6%)		193 (95.1%)	218 (92.4%)	
Soft tissues						
Yes (n = 114)	82 (8.9%)	32 (6.0%)	0.049	13 (6.4%)	16 (6.8%)	0.875
No (n = 1336)	837 (91.1%)	499 (94.0%)		190 (93.6%)	220 (93.2%)	

**Table 2 cancers-17-01833-t002:** *PIK3CA* mutation frequency in different regions of Russia.

Federal District	PIK3CA WT (n = 1179)	PIK3CA MUT (n = 693)	*p*-Value (PIK3CA WT vs. MUT)	PIK3CA MUT (Exon 10, n = 257)	PIK3CA MUT (Exon 20, n = 323)	*p*-Value (Exon 10 vs. Exon 20 Mutations)
Northwestern (n = 713)	451 (63.3%)	262 (36.7%)	0.800	103 (46.0%)	121 (54.0%)	0.400
Southern (n = 217)	133 (61.3%)	84 (38.7%)		29 (42.0%)	40 (58.0%)	
Central (n = 185)	110 (59.5%)	75 (40.5%)		33 (52.4%)	30 (47.6%)	
North Caucasian (n = 175)	116 (66.3%)	59 (33.7%)		27 (49.1%)	28 (50.9%)	
Volga (n = 129)	81 (62.8%)	48 (37.2%)		14 (33.3%)	28 (66.7%)	
Khanty-Mansi (n = 113)	78 (69.0%)	35 (31.0%)		9 (32.1%)	19 (67.9%)	
Siberian (n = 109)	67 (61.5%)	42 (38.5%)		17 (51.5%)	16 (48.4%)	
Urals (n = 88)	54 (61.4%)	34 (38.6%)		11 (45.8%)	13 (54.2%)	
Far Eastern (n = 88)	52 (59.1%)	36 (40.9%)		8 (29.6%)	19 (70.4%)	
Yamalo-Nenets (n = 13)	6 (46.2%)	7 (53.8%)		2 (33.3%)	4 (66.7%)	
ND (n = 42)						

**Table 3 cancers-17-01833-t003:** Univariable and multivariable analysis of the factors associated with event-free survival.

Parameter	Univariable		Multivariable	
	Hazard Ratio [95% CI]	*p*-Value	Hazard Ratio [95% CI]	*p*-Value
Age (<60 vs. ≥60 years) (n = 415)	1.473 [1.151–1.886] (≥60)	0.0006	1.482 [1.178–1.866]	0.0008
Grade (G1–2 vs. G3) (n = 216)	1.126 [0.818–1.550] (G3)	0.384		
Ki67 (<20% vs. ≥20%) (n = 342)	1.075 [0.860–1.344] (≥20%)	0.529		
Stage (I–III) (n = 431)	1.401 [1.213–1.619] *	<0.0001		
Tumor size (T1–2 vs. T3–4) (n = 412)	1.378 [1.068–1.777] (T3–4)	0.0063	1.301 [1.024–1.653]	0.031
Nodal involvement (N0 vs. N1–3) (n = 414)	0.687 [0.562–0.838] (N0)	0.0005	0.722 [0.577–0.903]	0.004
PR status (positive vs. negative) (n = 418)	1.020 [0.822–1.266] (PR-)	0.856		
PIK3CA (wt vs. mut) (n = 431)	1.055 [0.867–1.283] (mut)	0.592		

* Univariable analysis performed by Cox regression.

**Table 4 cancers-17-01833-t004:** Frequency of *PIK3CA* mutations in patients with various sensitivities to adjuvant endocrine therapy.

Type of Adjuvant Therapy	Primary Resistance	Secondary Resistance	Sensitivity	*p*-Value
Tamoxifen (n = 331)	66/163 (40.5%)	28/103 (27.2%)	28/65 (43.1%)	0.046
Aromatase inhibitors (n = 82)	19/42 (45.2%)	11/28 (39.3%)	4/12 (33.3%)	0.730

## Data Availability

The original contributions presented in this study are included in the article/Appendix A. Further enquiries can be directed to the corresponding author.

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
