# Peer review of "Clinical and Morphological Features of ER-Positive HER2-Negative Breast Tumors with PIK3CA Mutations in Russian Patients"

_cancers, 2025, doi:10.3390/cancers17111833_

Round 1
Reviewer 1 Report
Comments and Suggestions for Authors
Mutations in the PIC3CA gene are one of the most common genetic events in the carcinogenesis of breast cancer and meanwhile can be targeted therapeutically through various PIC3CA inhibitors. Although the signaling function of PIC3CA is well understood, data on the clinical relevance of these mutations and their correlation with the prognosis are still heterogeneous. Many studies suggest that PIC3CA mutations exert a negative influence on the patients’ prognosis and survival but opposite data also exist.
In this context, a systematic study of a large cohort in one big country (Russia) focusing at PIC3CA mutations is principally of interest. The present study involved 1872 patients recruited from multiple institutions from various regions throughout Russia. The patients included were restricted to hormone receptor-positive and HER2-negative cases rendering the cohort more homogeneous. Tumor tissue was subjected to a stepwise combination of molecular biological methods involving high-resolution melting analysis up to pyrosequencing. In this manner, a higher sensitivity of mutation detection as compared to standard commercial kits was obtained. Various statistical correlations were determined. Interesting results include the association of PIC3CA mutations with several prognostically favorable parameters such as lower histological grade and lower proliferation, although in survival analyses there was no effect on event-free survival. There was also no influence of different ethnicities within Russia.
In general, this is an interesting large-scale molecular biological analysis of a gene with high tumor biological and clinical importance in breast cancer. The results are appropriately discussed in relation to other studies. The paper is clearly presented and contains all relevant literature.
There are a few points which should be considered by the authors:
- The restriction to HR-positive and HER2-negative cases is a central feature of this study and should therefore already be included in the title. The current form of the title ("…breast tumors with PIK3CA mutations…") misleadingly suggests the inclusion of breast tumors of any type.
- Line 211: Why "advanced" breast cancer? Not all cases are advanced, the study population includes 184 T1 cases.
- Table 1, section "Primary tumor size", line T4: What does the P-value "0" mean?
- Line 289, Table 3 etc.: The authors consistently use the statistical terms "univariable" and "multivariable" analysis. It should be checked whether these are the correct terms or whether rather the terms "univariate" and "multivariate" are appropriate in this context.
Author Response
Comment 1: The restriction to HR-positive and HER2-negative cases is a central feature of this study and should therefore already be included in the title. The current form of the title ("…breast tumors with PIK3CA mutations…") misleadingly suggests the inclusion of breast tumors of any type.
Response: Thank you for pointing this out. We have changed the title to “Clinical and Morphological Features of ER-positive HER2-negative Breast Tumors with PIK3CA Mutations in Russian Patients”
Comment 2: Line 211: Why "advanced" breast cancer? Not all cases are advanced, the study population includes 184 T1 cases.
Response: We agree with this comment and have deleted the word “advanced” from the corresponding sentence in the introduction.
Comment 3: Table 1, section "Primary tumor size", line T4: What does the P-value "0" mean?
Response: Thank you for pointing this out. It was a typing error, now it has been deleted.
Comment 4: Line 289, Table 3 etc.: The authors consistently use the statistical terms "univariable" and "multivariable" analysis. It should be checked whether these are the correct terms or whether rather the terms "univariate" and "multivariate" are appropriate in this context.
Response: We acknowledge that the terms 'univariate' and 'univariable', and 'multivariate' and 'multivariable', are often used interchangeably in the literature. However, they represent distinct methodological approaches [1]. For example, while the multivariable model is used for the analysis with one outcome (dependent) and multiple independent (predictor or explanatory) variables, multivariate is used for the analysis with more than 1 outcome (e.g. repeated measures) and multiple independent variables. Therefore, we can conclude that in our manuscript the usage of these terms is correct.
- Ebrahimi Kalan M, Jebai R, Zarafshan E, Bursac Z. Distinction Between Two Statistical Terms: Multivariable and Multivariate Logistic Regression. Nicotine Tob Res. 2021 Aug 4;23(8):1446-1447. doi: 10.1093/ntr/ntaa055. PMID: 32215638.
Reviewer 2 Report
Comments and Suggestions for Authors
The study conducted a comprehensive PIK3CA mutational analysis on tumor samples from 1872 Russian HR+/HER2– BC patients, encompassing 542 mutation types. This thorough examination offers profound insights into the PIK3CA mutational landscape in Russian BC patients. The research identified 5.5–12% of PIK3CA mutations that are potentially undetected by current commercial assays. These novel mutations broaden the recognized mutational spectrum of the PIK3CA gene and hold significant implications for future therapeutic interventions and research. The occurrence of PIK3CA mutations was correlated with favorable clinical traits such as older patient age, reduced tumor size and grade, decreased Ki67 positivity rate, and PR status positivity. These findings pave the way for tailored treatment selections for patients. The results underscore the necessity of expanded testing beyond conventional hotspots in the PIK3CA gene to ensure no potential actionable targets are overlooked, guiding principles for forthcoming personalized treatment approaches. The study presents a wealth of clinical data, encompassing patient age, disease stage at diagnosis, tumor dimensions, lymph node involvement, and distant metastases. This information forms a robust foundation for exploring the correlation between PIK3CA mutations and disease trajectory. This manuscript is acceptable with minor revisions. The following are the revision suggestions.
- Comprehensive functional and mechanistic analyses of recently identified PIK3CA mutations are elucidating their significance in cancer pathogenesis.
- It is suggested that studies incorporate data from inhibitors of the PI3K/AKT/mTOR pathway that have recently gained approval or are currently undergoing clinical trials.
- The following literature is closely related to the author's topic. It is recommended that the author cite it.
[1] A. Gu, J. Li, M.-Y. Li, Y. Liu, Patient-derived xenograft model in cancer: establishment and applications. MedComm, 2025, 6, e70059. DOI: 10.1002/mco2.70059
[2] Yuan H. Clinical decision making: Evolving from the hypothetico-deductive model to knowledge-enhanced machine learning. Med Adv. 2024; 2(4): 375–379. https://doi.org/10.1002/med4.83
[3] P. Ma, G. Wang, K. Men, et al. Advances in clinical application of nanoparticle-based therapy for cancer treatment: A systematic review, Nano TransMed. 3 (2024) 100036. https://doi.org/https://doi.org/10.1016/j.ntm.2024.100036
- In the evaluation of clinical characteristics, it is imperative to account for and control a broader range of potential confounding variables. For instance, factors such as the degree of obesity, lifestyle, and other elements that could influence the outcome should be adjusted to ensure the precision of the analysis.
Author Response
Comment 1: Comprehensive functional and mechanistic analyses of recently identified PIK3CA mutations are elucidating their significance in cancer pathogenesis.
Response: We fully agree that functional studies are invaluable for characterizing the consequences and significance of specific mutations. We have added the discussion on the functional roles of the identified variants (lines 351-357):
"Most of the identified PIK3CA alterations (37 out of 43 mutation types and 731 out of a total of 739 mutations) have been reported previously. However, we could not find any mention of several of the identified variants (p.P124S, p.H419_C420delinsR, p.H419_P421>QT, p.C420_P421del, p.N1044H and p.H1048A) in the literature. There is experimental evidence of the activating role of the majority of the found alterations (30/43 mutation types; 723/739 individual mutations) [28,41-46]. Nevertheless, the functional significance of the newly identified mutation types remains to be evaluated."
- Gymnopoulos M, Elsliger MA, Vogt PK. Rare cancer-specific mutations in PIK3CA show gain of function. Proc Natl Acad Sci U S A. 2007 Mar 27;104(13):5569-74. doi: 10.1073/pnas.0701005104.
- Jin N, Keam B, Cho J, Lee MJ, Kim HR, Torosyan H, Jura N, Ng PK, Mills GB, Li H, Zeng Y, Barbash Z, Tarcic G, Kang H, Bauman JE, Kim MO, VanLandingham NK, Swaney DL, Krogan NJ, Johnson DE, Grandis JR. Therapeutic implications of activating noncanonical PIK3CA mutations in head and neck squamous cell carcinoma. J Clin Invest. 2021 Nov 15;131(22):e150335. doi: 10.1172/JCI150335.
- Ikenoue T, Kanai F, Hikiba Y, Obata T, Tanaka Y, Imamura J, Ohta M, Jazag A, Guleng B, Tateishi K, Asaoka Y, Matsumura M, Kawabe T, Omata M. Functional analysis of PIK3CA gene mutations in human colorectal cancer. Cancer Res. 2005 Jun 1;65(11):4562-7. doi: 10.1158/0008-5472.CAN-04-4114.
- Ng PK, Li J, Jeong KJ, Shao S, Chen H, Tsang YH, Sengupta S, Wang Z, Bhavana VH, Tran R, Soewito S, Minussi DC, Moreno D, Kong K, Dogruluk T, Lu H, Gao J, Tokheim C, Zhou DC, Johnson AM, Zeng J, Ip CKM, Ju Z, Wester M, Yu S, Li Y, Vellano CP, Schultz N, Karchin R, Ding L, Lu Y, Cheung LWT, Chen K, Shaw KR, Meric-Bernstam F, Scott KL, Yi S, Sahni N, Liang H, Mills GB. Systematic Functional Annotation of Somatic Mutations in Cancer. Cancer Cell. 2018 Mar 12;33(3):450-462.e10. doi: 10.1016/j.ccell.2018.01.021.
- Oda K, Okada J, Timmerman L, Rodriguez-Viciana P, Stokoe D, Shoji K, Taketani Y, Kuramoto H, Knight ZA, Shokat KM, McCormick F. PIK3CA cooperates with other phosphatidylinositol 3'-kinase pathway mutations to effect oncogenic transformation. Cancer Res. 2008 Oct 1;68(19):8127-36. doi: 10.1158/0008-5472.CAN-08-0755.
- Miled N, Yan Y, Hon WC, Perisic O, Zvelebil M, Inbar Y, Schneidman-Duhovny D, Wolfson HJ, Backer JM, Williams RL. Mechanism of two classes of cancer mutations in the phosphoinositide 3-kinase catalytic subunit. Science. 2007 Jul 13;317(5835):239-42. doi: 10.1126/science.
Comment 2: It is suggested that studies incorporate data from inhibitors of the PI3K/AKT/mTOR pathway that have recently gained approval or are currently undergoing clinical trials.
Response: We have extended the part of the introduction devoted to various PI3K/AKT inhibitors (lines 174-184):
“Monotherapy of solid neoplasms with PI3K inhibitors proved ineffective, however, several compounds have already been approved for the treatment of PIK3CA-mutated hormone-receptor (HR)-positive/HER2-negative breast cancer (BC) in combination with other agents. Combined use of selective inhibitor of the PI3K-alpha alpelisib and estrogen degrader fulvestrant received an approval for patients progressing on or after prior endocrine-based therapy [2,3]. Similar approval was granted to AKT inhibitor capivasertib, however, in addition to PIK3CA mutations, the indication of this drug includes nucleotide sequence alterations in AKT1 and PTEN genes [4]. Another AKT inhibitor, ipatasertib, is currently under investigation in breast cancer [5]. Inavolisib has an ability to drive the degradation of mutated p110-alpha while sparing the wild-type isoform of this protein; it has been approved for clinical use in combination with fulvestrant and CDK4/6 inhibitor palbociclib [6]. Several other PI3K inhibitors have failed to achieve a significant improvement in progression-free survival or the objective response rate, and their use has been associated with considerable toxicity (e.g. buparlisib, pictilisib and taselisib). A number of novel compounds, including inhibitors that selectively target mutant PIK3CA isoforms, have shown promising results in preclinical and early-phase clinical studies [5]. There are also ongoing trials utilizing combinations of PI3K inhibitors with other drugs and involving other than BC cancer types [7-10].”
- Hao C, Wei Y, Meng W, Zhang J, Yang X. PI3K/AKT/mTOR inhibitors for hormone receptor-positive advanced breast cancer. Cancer Treat Rev. 2025 Jan;132:102861. doi: 10.1016/j.ctrv.2024.102861. Epub 2024 Nov 19. PMID: 39662202.
Comment 3: The following literature is closely related to the author's topic. It is recommended that the author cite it.
[1] A. Gu, J. Li, M.-Y. Li, Y. Liu, Patient-derived xenograft model in cancer: establishment and applications. MedComm, 2025, 6, e70059. DOI: 10.1002/mco2.70059
[2] Yuan H. Clinical decision making: Evolving from the hypothetico-deductive model to knowledge-enhanced machine learning. Med Adv. 2024; 2(4): 375–379. https://doi.org/10.1002/med4.83
[3] P. Ma, G. Wang, K. Men, et al. Advances in clinical application of nanoparticle-based therapy for cancer treatment: A systematic review, Nano TransMed. 3 (2024) 100036. https://doi.org/https://doi.org/10.1016/j.ntm.2024.100036
Response: The three publications mentioned cover various aspects of experimental and clinical oncology. However, they do not appear to be directly related to the study of somatic PIK3CA mutations presented in our manuscript. Therefore, we consider it unreasonable to incorporate them into our article.
Comment 4: In the evaluation of clinical characteristics, it is imperative to account for and control a broader range of potential confounding variables. For instance, factors such as the degree of obesity, lifestyle, and other elements that could influence the outcome should be adjusted to ensure the precision of the analysis.
Response: We agree that including additional factors would improve the accuracy of the analysis. Unfortunately, we do not have the necessary data or access to information on various lifestyle factors for our sample.